# Microgeographic Wing-Shape Variation in *Aedes albopictus* and *Aedes scapularis* (Diptera: Culicidae) Populations

**DOI:** 10.3390/insects11120862

**Published:** 2020-12-03

**Authors:** Rafael Oliveira-Christe, André Barretto Bruno Wilke, Mauro Toledo Marrelli

**Affiliations:** 1Institute of Tropical Medicine, University of São Paulo, Av. Dr. Enéas Carvalho de Aguiar, 470, 05403-000 Butanta, SP, Brazil; rchriste@usp.br; 2Department of Public Health Sciences, Miller School of Medicine, University of Miami, Miami, FL 33136, USA; axb1737@med.miami.edu; 3Department of Epidemiology, School of Public Health, University of São Paulo, Av. Dr. Arnaldo, 715, 01246-904 Butanta, SP, Brazil

**Keywords:** microevolution, Culicidae, mosquitoes, urbanization, urban parks

## Abstract

**Simple Summary:**

*Aedes albopictus* and *Aedes scapularis* have been incriminated as vectors of arboviruses that can cause human diseases. Geometric morphometric tools have been used in several epidemiological studies to investigate how each of these mosquito species behaves in urban areas in the city of São Paulo, Brazil, and how these species have adapted to anthropogenic changes in the environment. Since it is exotic to the Brazilian fauna, *Ae. albopictus* has received more attention from health agencies than *Ae. scapularis*, a native species. It is thus crucial to investigate and compare the two species simultaneously in the same geographic area to better understand how they adapt to urban environments. The aim of this work was to evaluate the population profile of these species in urban parks in the city of São Paulo using wing geometric morphometrics. Our results showed different levels of population structuring for both species, suggesting different adaptive responses to urbanization: *Ae. albopictus* populations collected in the urban parks displayed homogeneous wing patterns, whereas *Ae. scapularis* populations were shown to have more variation. This indicates the importance of maintaining surveillance of exotic and native mosquito vector species given the fundamental role that urbanization can play in the population dynamics of arbovirus vector species.

**Abstract:**

*Aedes albopictus* and *Aedes scapularis* are vectors of several arboviruses, including the dengue, chikungunya, and Rocio virus infection. While *Ae. albopictus* is a highly invasive species native to Asia and has been dispersed by humans to most parts of the world, *Ae. scapularis* is native to Brazil and is widely distributed in the southeast of the country. Both species are highly anthropophilic and are often abundant in places with high human population densities. Because of the great epidemiological importance of these two mosquitoes and the paucity of knowledge on how they have adapted to different urban built environments, we investigated the microgeographic population structure of these vector species in the city of São Paulo, Brazil, using wing geometric morphometrics. Females of *Ae. albopictus* and *Ae. scapularis* were collected in seven urban parks in the city. The right wings of the specimens were removed and digitized, and eighteen landmarks based on vein intersections in the wing venation patterns were used to assess cross-sectional variation in wing shape and size. The analyses revealed distinct results for *Ae. albopictus* and *Ae. scapularis* populations. While the former had less wing shape variation, the latter had more heterogeneity, indicating a higher degree of intraspecific variation. Our results indicate that microgeographic selective pressures exerted by different urban built environments have a distinct effect on wing shape patterns in the populations of these two mosquito species studied here.

## 1. Introduction

The genus *Aedes* (Diptera: Culicidae) has species that are competent vectors of human pathogens [1]. Besides *Aedes aegypti* Linnaeus, 1792, the main vector of dengue virus, several other *Aedes* species have been incriminated in the transmission of arboviruses [2,3,4,5], including *Ae. albopictus* Skuse, 1894, and *Ae. scapularis* Rondani, 1848, the subjects of this study.

The Asian tiger mosquito (*Ae. albopictus)*, a species native to Southeast Asia [6] found in sylvatic, urban and rural areas [7], can transmit dengue, Zika, yellow fever and chikungunya viruses [8,9]. Because it can survive in artificial containers for months, this species has spread around the world to Africa, the Middle East, Europe and the Americas [6,10,11,12], including Brazil [3], as a result of international trade and the transportation of goods, particularly tires [10,13,14].

*Aedes scapularis* is a Neotropical mosquito species native to Brazil that can be found in large cities, including São Paulo [3,15,16]. It is commonly found in abundance in remnants of the Atlantic Forest, as well as rural, peri-urban and urban areas [3,17]. The species was incriminated as a vector of the Rocio virus during an outbreak in the southwest of the state of São Paulo between 1974 and 1978 [18].

In most large cities, vegetation tends to be restricted to areas known as “green islands” or parks, whose main characteristic is their fragmented arrangement along the edge of the urban area [19] and which are often used by the local community for outdoor activities, such as sports, and for recreational purposes as well as to enjoy nature. They not only constitute areas of preserved wilderness but also serve as a refuge for endemic and exotic animal and plant species [20,21] and can favor the establishment of adapted mosquito species, which rapidly respond to the heterogeneous dynamics of these habitats [22,23,24].

Some studies have shown that an increase in human changes to the environment can promote biotic homogenization, in turn increasing contact between pathogens, human hosts and mosquito vectors [25,26]. This scenario greatly increases the risk of arbovirus transmission, directly impacting public health [27,28]. Urban areas are at increased risk of arbovirus transmission as they may harbor vector mosquito species that are adapted to and thrive in urbanized ecosystems [29,30]. Given that urbanization has affected both native and exotic mosquito species, as demonstrated by previous studies [16,31,32,33,34], understanding how different vector species cope and adapt in the face of temporal and spatial variations in the environment is crucial to elucidate their population dynamics and disease transmission patterns [33,35].

Wing geometric morphometrics has proved to be a useful tool for studying size and shape variations in mosquito wings [36]. Anatomical landmarks based on wing venation provide reliable information, making it possible to identify and distinguish sibling species [37], cryptic species and species complexes [38,39,40], as well as to investigate sexual dimorphism [41,42] and mosquito population structure [43,44,45]. Based on previous studies by our group using wing geometric morphometrics [34,45,46], we hypothesized that under similar environmental and temporal conditions, populations of *Ae. scapularis* and *Ae. albopictus*, native and invasive mosquito species, respectively, in Brazil, have been affected differently by urbanization processes. We used wing geometric morphometrics to investigate variations in wing size and shape among populations of both species in urban parks in the city of São Paulo.

## 2. Materials and Methods

### 2.1. Mosquito Collections

Mosquito collections were performed between 2011 and 2013 in the following urban parks inserted in highly urbanized areas of the city of São Paulo: Anhanguera (ANH), Alfredo Volpi (ALV), Burle Marx (BLM), Piqueri (PQR), Previdencia (PRV), Santo Dias (STD) and Shangrilá (SHL) (Figure 1, Table 1). These parks were selected because of their convenient location near congested, highly urbanized areas and the fact that collections could be performed without unnecessary risk [22]. To avoid collecting sibling specimens, the specimens were randomly selected, and different collection techniques were used: manual 12 V aspirators [47], CO_2_-baited CDC light traps and Shannon traps [48].

All the parks in the study have similar patterns of vegetation and are composed of secondary forest, except for Shangrilá Park, which is next to an environmental protection area [45]. Sympatry of *Ae. scapularis* and *Ae. albopictus* was observed in three of the seven parks: AFV, BLM and PQR. All the specimens were conditioned and sent to the Public Health Entomology Laboratory at the University of São Paulo, where they were identified with the taxonomic key by Consoli and Lourenço-de-Oliveira [49].

### 2.2. Wing Preparation

The right wing was removed from each female, mounted between a slide and coverslip with Canada balsam (Sigma-Aldrich, St. Louis, MO, USA) and photographed in a Leica M205C stereoscope under 40× magnification. Eighteen landmarks at intersections in the wing venation patterns were then digitized with TpsDig software 1.4 [50] as in Christe et al. [42] and Carvalho et al. [45].

### 2.3. Morphometric Analysis

To measure the isometric estimator (Centroid Size, CS) of each population, we used MorphoJ 1.02 [51]. The results of non-parametric ANOVA of CS with a *post hoc* Tukey test were analyzed in PAST 1.89 [45]. Multiple regression analysis of the Procrustes coordinates on CS was performed to estimate the allometric effect of wing size on wing shape with TpsUtil 1.29 and TpsRelw 1.39 [16]. To assess the effect of wing size on wing shape (allometry) we performed a multivariate regression of the Procrustes coordinates against CS using a permutation test with 10,000 randomizations in MorphoJ 1.02.

Cross-validated reclassification was carried out for each *Ae. albopictus* and *Ae. scapularis* specimen for all populations with MorphoJ 1.05 to evaluate the degree of dissimilarity between samples. Each mosquito was reclassified according to its wing similarity to the average shape of each group based on Mahalanobis distances to test the accuracy of morphometric analyses. To test for isolation by distance (IBD), we used the Mantel test in PAST 1.89 based on the geographic distances (linear kilometers) between collection sites and Mahalanobis distances [52]. Canonical Variate Analysis (CVA) was used to explore the degree of wing shape dissimilarity in all the *Ae. albopictus* and *Ae. scapularis* populations. Mahalanobis distances were calculated to determine the phenotypic distance between samples, and a Neighbor-Joining tree (NJ) was constructed using PAST 1.89. *Aedes aegypti* (*N* = 30) was used as an outgroup. Wireframe graphs were plotted in MorphoJ to compare the level of wing deformation between the *Ae. albopictus* and *Ae. scapularis* populations.

## 3. Results

Centroid size for the *Ae. scapularis* populations varied from 1.85 mm to 1.96 mm. The ALV population had the highest value (mean 1.95 mm), and the STD population the lowest (mean 1.92 mm) (Figure 2A). Mean CS for the *Ae. scapularis* populations differed significantly between the AFV and STD populations (ANOVA: F(5,12), *p* < 0.01) (Appendix A). Mean CS for the *Ae. albopictus* populations varied from 1.88 mm to 2.01 mm; the highest mean corresponded to the SHL population (1.96 mm), and the lowest to the PQR population (1.94 mm) (Figure 2B). No statistically significant differences in mean CS were found between the *Ae. albopictus* populations (ANOVA: F(3,498), *p* < 0.01) (Appendix A).

The influence of wing size on wing shape (allometry) was statistically significant (*p* < 0.0001) for both *Ae. scapularis* (5.03%) and *Ae. albopictus* (4.71%) populations and was thus removed in the subsequent analysis.

The results of the CVA differed with *Ae. scapularis* wing shape being more heterogeneous than *Ae. albopictus*. Among the former, the ALV population differed the most from all the others: it was completely segregated from the ANH and BLM populations and overlapped with PQR and STD only slightly. The *Ae. scapularis* populations that differed the most from each other were BLM and ALV (Figure 3A). Among the *Ae. albopictus* populations, ANH had higher levels of wing shape variation, with just a minor overlap with the PQR population and no similarities with the PRV population. The BLM population had the greatest degree of overlap with all the other populations, and the SHL population also overlapped with all the other populations, although to a lesser extent with the ANH population (Figure 3B). This result is in agreement with the ones obtained in the UPGMA phenogram based on Procrustes distances of *Ae. scapularis* and *Ae. albopictus* populations (Appendix A, Appendix A).

*Aedes scapularis* populations yielded higher levels of variation in the Mahalanobis distances, 1.6815 to 3.4416, when compared to *Ae. albopictus* populations (1.6163 to 2.6631). All *Ae. scapularis* comparisons were statistically significant (*p* < 0.05), the only exception being ANH vs. BLM (*p* = 0.1846) (Table 2). A similar result was obtained for *Ae. albopictus*, in which, except for SHL vs. PRV, all the other comparisons were statistically significant (*p* < 0.05) (Table 3).

In the NJ tree for all the *Ae. scapularis* populations, the ANH and BLM populations separated from the other populations and grouped together with a high bootstrap value (99%). The other *Ae. scapularis* populations grouped in branches with lower bootstrap values (51% for PQR, 51% for STD and 65% for ALV), indicating that some ecological features in ANH and BLM parks have led to changes in wing shape not observed in the populations of the other parks. 

In the NJ tree for all the *Ae. albopictus* populations, the ANH population segregated completely with a bootstrap value of 100%. The PRV and SHL populations clustered in the same branch with a bootstrap value of 97%. The PQR and BLM populations yielded low bootstrap values. These results may indicate that the unique characteristics of the ANH park influence wing shape patterns in the *Ae. albopictus* population differently and that the same process may be occurring in PRV and SHL parks (Figure 4).

Pairwise cross-validated reclassification of the *Ae. scapularis* populations resulted in high scores. In the group 1 vs. group 2 reclassification, the highest scores were AFV × ANH (88%), AFV × BLM (80.7%) and PQR × BLM (75%). In the group 2 vs. group 1 reclassification, the highest scores were ANH × ALV (78.5%), BLM × ALV (72%), PQR × BLM (76.1%) and STD × BLM (76%) (Table 4). These high scores indicate that there are substantial differences in wing shape patterns in the *Ae. scapularis* populations studied here.

The same analysis for *Ae. albopictus* revealed low levels of variation between populations, indicating homogeneous wing shape. In group 1 vs. group 2 comparisons, the values ranged from 45% (BLM × ANH) to 66.6% (PQR × BLM); no higher values were observed. In the group 2 vs. group 1 comparisons, ANH × PRV had a score of 81.4%, indicating considerable wing variation in these populations, while the other pairwise comparisons revealed moderate to little variation between populations (Table 5).

The results of the Mantel test failed to reveal a correlation between Mahalanobis distances and geographic distances for either species: *r =* 0.3292, *p = 0.968* for *Ae. scapularis;* and *r =* 0.1698, *p* = 0.125 for *Ae. albopictus*. This indicates that factors other than distance were responsible for the variation in wing shape patterns found in the mosquito populations studied here.

The wireframe graph shows the levels of variation in landmarks for *Ae. albopictus* and *Ae. scapularis.* In the latter, landmarks 1, 2, 10, 11 and 13 varied more than the remaining landmarks, whereas in the former all the landmarks showed some degree of variation, although landmarks 1, 2, 3, 8, 9, 10, 12, 14 and 15 varied more (Figure 5).

Similar results were obtained when performing all analyses described here considering only the population from the three parks where both species were sympatric (ANH, BLM, PQR). *Aedes albopictus* populations showed a homogeneous wing pattern, whereas *Ae. scapularis* populations showed higher levels of wing shape variation.

## 4. Discussion

Urbanization results in a range of environmental impacts, including an increase in temperature, loss of habitat, pollution, and deforestation [53]. Anthropogenic changes to the environment often benefit exotic mosquito species that can exploit the available resources and thrive in human-modified habitats [54]. As one of the effects of the urbanization process, the urban parks of São Paulo city are considered “green islands”, which serve as shelter and refuge for many native and invasive species, and the loss of vegetal cover is one effect of this process [22,34].

Our results showed that wing shape in populations of the native species *Ae. scapularis* and the invasive species *Ae. albopictus* found sympatrically in urban parks in the city of São Paulo and therefore under the same environmental conditions was affected differently by urbanization. These sympatric populations exhibited different degrees of wing shape and size variation. *Aedes albopictus* yielded lower values in the cross-validated classification test when compared to *Ae. scapularis*. This result indicates that selective pressures in the urban environments here have affected these species differently, supporting the findings of previous studies [55,56]. However, further studies are needed to allow a deeper understanding of this phenomenon. Our results showed that while *Ae. scapularis* populations were segregated into two completely distinct clusters in the NJ tree based on Mahalanobis distances, the *Ae. albopictus* populations showed higher levels of overlapping, with no great distinction between them. The variation found in wing size between populations of *Ae. scapularis* and *Ae. albopictus* might be associated with intrinsic characteristics of the breeding sites, such as temperature, physicochemical parameters of the water, and food availability [57].

The finding of wing morphometric variability in the *Ae. scapularis* populations using CVA, cross-validation and the NJ tree is significant as the mosquitoes were collected no more than 50 km apart. Although IBD did not show a positive correlation, other factors may be modulating the microgeographic structure of these mosquito populations.

The *Ae. scapularis* populations from ALV had wing shape patterns substantially different from those of the BLM and ANH populations. This difference can be attributed to different levels of urbanization in these green areas [45,46]. BLM and ANH parks each have a large conserved area [57] that can provide a greater supply of potential breeding sites, sugar sources and resting places than the ALV park.

The lack of heterogeneity revealed by CVA in the *Ae. albopictus* populations, of which only the ANH and PQR populations showed some level of segregation, is in agreement with the results of Vidal et al. [31], although in their study segregation between populations, which was analyzed over four years, was investigated in relation to time rather than distance and environmental variables as in the present study.

The partial segregation of the ANH *Ae. albopictus* population in the CVA was confirmed in the NJ tree, in which the population was segregated in an independent branch with a high (100%) bootstrap value. In the cross-validated reclassification, this population had a reclassification rate of 81.4% compared with the PQR population, confirming the results of the CVA. A possible explanation for the segregation of the ANH population is that this park is a protected area with native fauna and flora. This unique environment may have been responsible for the distinct wing shape pattern found in *Ae. albopictus* mosquitoes collected in this area [57]. This phenomenon of wing shape variation influenced by changes in the natural habitat of mosquitoes has been reported in *Anopheles cruzii* from natural environments and urbanized areas [34]. ANH Park is the largest remnant of forested area in the city, and the collection site, located in a transition area between the more conserved, wooded part of the park, to which the public do not have access, and the urban part, can provide suitable conditions for *Ae. albopictus* [22] to survive and maintain a homogeneous wing shape pattern, unlike the populations from the other parks. Although, some studies using landmark-based wing geometric morphometric to investigate intraspecific variation in mosquito populations have suggested that wing shape heterogeneity may be associated with the proximity of the collection sites [45,58], the finding that IBD was not significant for the *Ae. albopictus* and *Ae. scapularis* populations indicate that factors other than distance, such as local environmental factors, have a higher association with wing variation.

*Aedes albopictus* larval habitats are more numerous in urban areas [59], where the environmental conditions allow immature stages to develop faster and favor greater survival of all stages of the species, from larva to adult, unlike in rural areas [60]. This contrasts with the behavior of species native to tropical zones, such as *Ae. scapularis* [3], which have a tendency to colonize larger water collections [60]. *Aedes albopictus* can exploit breeding sites with a wide variety of sizes and characteristics, making it difficult to locate and eliminate potential breeding sites [59].

Zouhouli et al. [61] reported that native *Aedes* species are more frequently associated with natural breeding sites found at ground level, which may explain the finding by Medeiros-Sousa et al. [62] of more immature forms of *Ae. albopictus* than of *Ae. scapularis* in the ANH park. In their study, there were substantially more *Ae. albopictus* immatures in bamboo stumps and artificial containers than *Ae. scapularis* immatures, which were restricted to ponds and puddles. This finding is supported by the wide ecological valence and phenotypic plasticity of exotic species, leading to increased survival as a result of a decrease in the number of predators or competitors [63]. The wireframe analyses in the present study corroborate this, as they revealed a greater variation in wing landmarks in *Ae. albopictus*, an exotic mosquito in Brazil than in *Ae. scapularis*, a native species. The high wing shape variability found in *Ae. scapularis* populations in the present study can be attributed to the different selective pressures in each park, which may result in the adaptation of wing shape. Previous studies reported similar results for the same species using a morphometric approach [16].

The results of the CVA in the present study showed that the invasive mosquito *Ae. albopictus* has less morphological variance than the native *Ae scapularis*, suggesting different forms of morphological variation in response to habitat fragmentation. Vidal et al. [31] used a wing morphometric tool to demonstrate the temporal variation of *Ae. albopictus* populations in the city of São Paulo. In our study, both species had high levels of intraspecific variation in Mahalanobis distances as observed in the NJ trees. This may be explained by the different colonization histories of these species: while *Ae. scapularis* responded to fragmentation of natural habitat, *Ae. albopictus* colonized and spread to many regions of Brazil after first being reported in the country in 1986 and is now found in urban green areas and peri-urban areas [3].

The effects of invasive insect species on native insect species have been studied and documented according to the roles they play in their ecosystem, i.e., predator, prey, herbivore and detritivore [64]. In the short term, the presence of exotic species causes the displacement of native species, whose numbers decrease when they change their ecological niche, as shown in a previous study with ants and [64] carrion flies [65]. As reported by Zittra et al. [35], invasive species increase in abundance in a short period, making it fundamental to perform surveillance in aquatic habitats in order to determine whether exotic species have been colonizing breeding sites and to evaluate how invasion by these species can affect local native mosquito fauna.

## 5. Conclusions

Investigation of wing shape and size variations in *Ae. scapularis* and *Ae. albopictus* populations under similar environmental and temporal conditions in urban parks in the city of São Paulo showed that these species were affected differently by urbanization processes. *Aedes albopictus* populations yielded homogeneous wing patterns, whereas *Ae. scapularis* populations yielded higher levels of variation.

## Figures and Tables

**Figure 1 insects-11-00862-f001:**
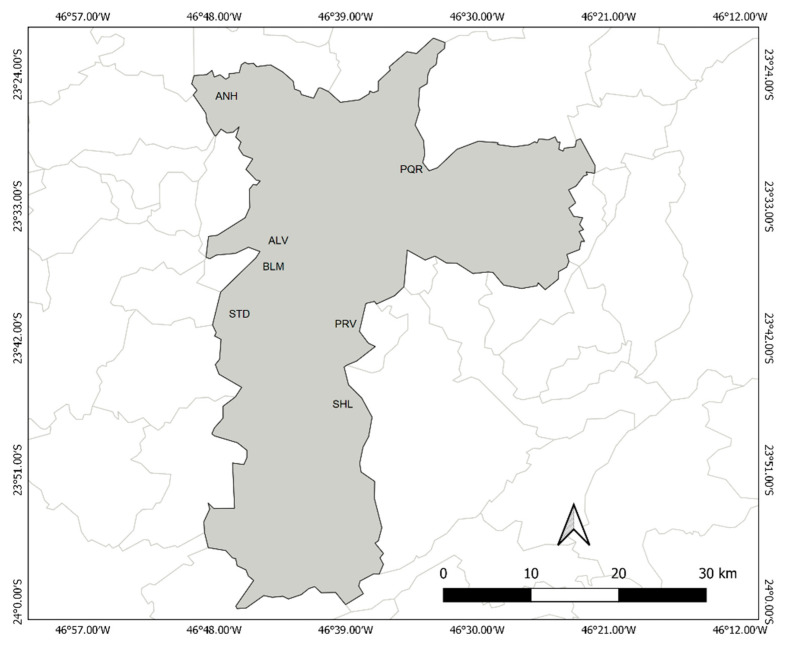
Locations where *Ae. albopictus* and *Ae. scapularis* sampling was performed in São Paulo, Brazil: Alfredo Volpi Park (ALV); Anhanguera Park (ANH); Burle Marx Park (BLM); Piqueri Park (PQR); Previdencia Park (PRV); Shangrilá Park (SHL); and Santo Dias Park (STD).

**Figure 2 insects-11-00862-f002:**
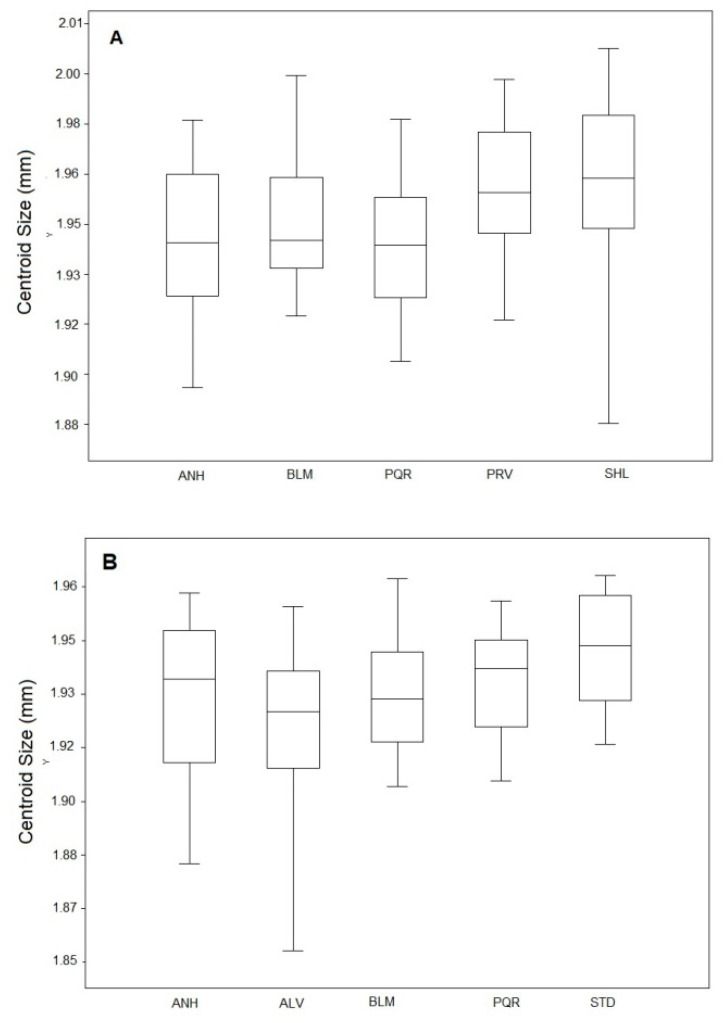
Boxplot graph showing differences in wing centroid size between *Ae.*
*scapularis* (**A**) and *Ae. albopictus* (**B**) populations.

**Figure 3 insects-11-00862-f003:**
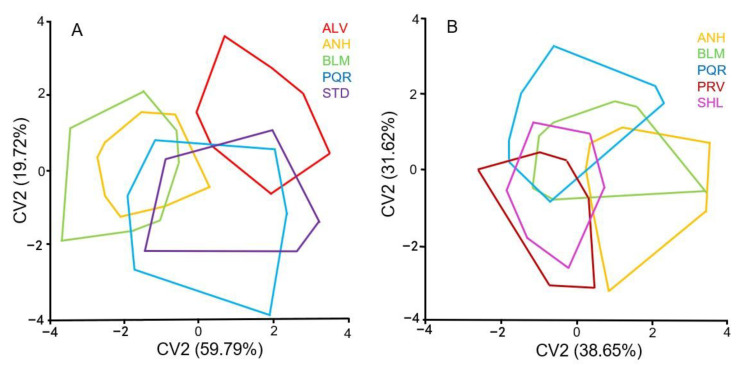
Morphospace produced by CVA of the wing shape of *Ae. scapularis* (**A**) and *Ae. albopictus* (**B**) populations.

**Figure 4 insects-11-00862-f004:**
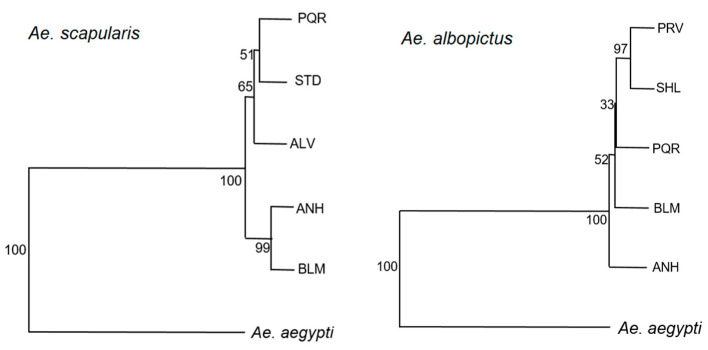
Neighbor-Joining tree based on Mahalanobis distances for *Ae. scapularis* and *Ae. albopictus* populations.

**Figure 5 insects-11-00862-f005:**
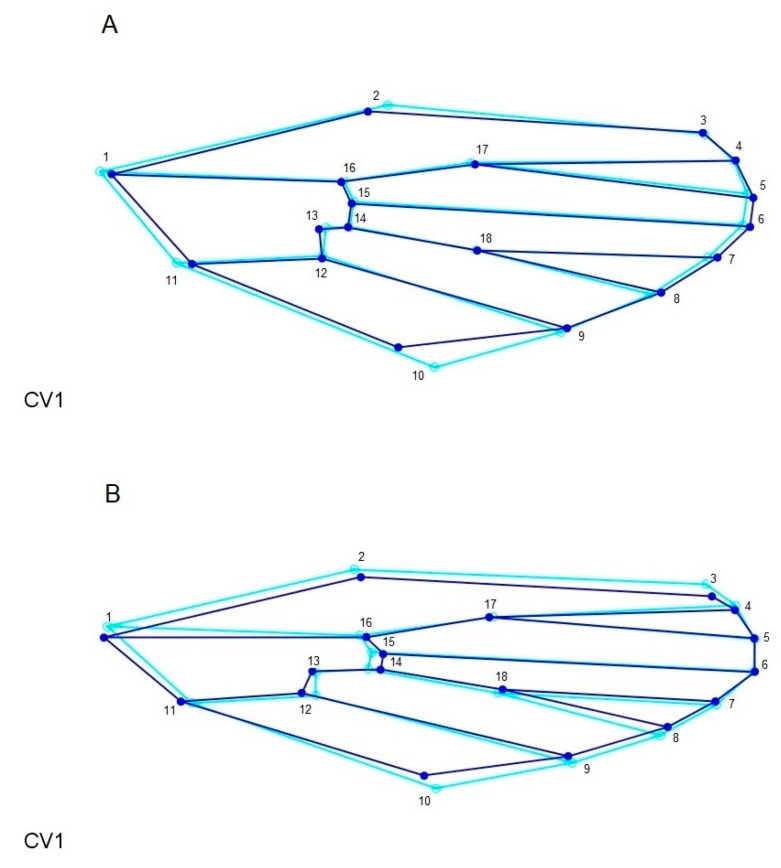
Superimposed wireframe graphs of *Aedes scapularis* (**A**) and *Aedes albopictus* (**B**). The light blue lines represent the medium wing shape variation for all populations and the dark blue lines represent the shape variance based on CV1.

**Table 1 insects-11-00862-t001:** *Aedes scapularis* and *Ae. albopictus* collection sites and data (- = no data).

Collection Site	Area (m^2^)	Plant Cover (%)	No. of Visitors Per Month	Coordinates	*Ae. scapularis* (N)	*Ae. albopictus* (N)	Collection Year
ALV	142,400	51.11	-	23°58′79.11″ S 46°70′27.80″ W	26	-	2011–2012
ANH	9,500,000	69.07	4000	23°29′33.36″ S 46°45′43.50″ W	28	20	2011–2013
BLM	138,279	29.12	18,000	23°37′55.92″ S 46°43′17.25″ W	25	15	2012–2013
PQR	97,200	7.33	25,000	23°31′39.98″ S 46°34′24.98″ W	23	24	2012–2013
PRV	91,500	51.77	-	23°34′40.99″ S 46°43′37.92″ W	-	28	2012–2013
STD	134,000	29.12	-	23°45′29.35″ S 46°46′23.18″ W	25	-	2011–2012
SHL	75,000	44.06	15,000	23°45′29.35″ S 46°39′44.28″ W	-	18	2011–2012

**Table 2 insects-11-00862-t002:** Mahalanobis distances (below diagonal) and *p* values (above diagonal) for *Ae. scapularis* populations.

Parks	ALV	ANH	BLM	PQR	STD
**ALV**	-	<0.0001	<0.0001	<0.0001	<0.0001
**ANH**	3.2379	-	0.1846	<0.0001	<0.0001
**BLM**	3.4416	1.6815	-	<0.0001	<0.0001
**PQR**	2.4979	2.8198	3.0182	-	0.0153
**STD**	2.3029	3.1487	3.3022	2.009	-

**Table 3 insects-11-00862-t003:** Mahalanobis distances (below diagonal) and *p* values (above diagonal) for *Ae. albopictus* populations.

Parks	ANH	BLM	PQR	PRV	SHL
**ANH**	-	0.0375	<0.0001	<0.0001	0.0141
**BLM**	2.3357	-	0.0099	0.0023	0.0011
**PQR**	2.6631	2.2677	-	<0.0001	0.0201
**PRV**	2.6218	2.3072	2.26	-	0.628
**SHL**	2.2551	2.5105	2.0854	1.6163	-

**Table 4 insects-11-00862-t004:** Pairwise cross-validated classification scores (%) for *Aedes scapularis* populations.

	**Group 2**					
**Group 1**	**Parks**	**ALV**	**ANH**	**BLM**	**PQR**	**STD**
	**ALV**	-	78.5	72	59.1	56
	**ANH**	88	-	56.5	59.1	66.6
	**BLM**	80.7	55.5	-	76.1	76
	**PQR**	48	74	75	-	52
	**STD**	53.8	74	68	56.5	-

**Table 5 insects-11-00862-t005:** Pairwise cross-validated classification scores (%) for *Ae. albopictus* populations.

	**Group 2**					
**Group 1**	**Parks**	**ANH**	**BLM**	**PQR**	**PRV**	**SHL**
	**ANH**	-	40	62.5	81.4	44.4
	**BLM**	45	-	62.5	67.8	61.1
	**PQR**	60	66.6	-	59.2	41.1
	**PRV**	47.3	60	54.5	-	62.5
	**SHL**	50	46.6	50	57.6	-

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
