# Peer review of "Microgeographic Wing-Shape Variation in *Aedes albopictus* and *Aedes scapularis* (Diptera: Culicidae) Populations"

_insects, 2020, doi:10.3390/insects11120862_

Round 1
Reviewer 1 Report
Review of the article
Rafael Oliveira-Christe, André Barretto Bruno Wilke and Mauro Toledo Marrelli
Microgeographic wing-shape variation in Aedes albopictus and Aedes scapularis (Diptera: Culicidae) populations
Study of microgeographic changes in wing shape of Aedes albopictus and Aedes scapularis (Diptera: Culicidae) populations is relevant due to their epidemiological significance. Analysis of the mosquito populations adaptation in an ever-changing urban environment is interesting. The authors have studied two species: Aedes albopictus and Ae. scapula, from 2011 to 2013. The results are interesting.
However, it was necessary to specify the gender composition of the analyzed mosquitoes, as they are well diagnosed by morphological characteristics. It is also necessary to indicate how many mosquitoes were collected each year and show their morphological analysis each year respectively. It is necessary because a seasonal variation in morphological characteristics can have a place. The identified morphological changes may also be related to the water temperature of larval stage development. High and low water temperatures have different effects to the larval development, its size, and, accordingly, can effect to the morphological parameters of the imago. Also, it was interesting to indicate the ratio of species in the sympatric zones, whether there is competition between the two species and its affect to the morphological variations of the two species Aedes albopictus and Ae. scapula.
Author Response
Dear Reviewer
Thank you very much for reviewing our manuscript and for the suggestions, which have enabled us to greatly improve the text.
We have made the changes to the manuscript to address the points, and we hope the article is now suitable for publication in Insects.
Sincerely,
Mauro Marrelli
Reviewer 1 Points:
Point 1: “It was necessary to specify the gender composition of the analyzed mosquitoes, as they are well diagnosed by morphological characteristics. It is also necessary to indicate how many mosquitoes were collected each year and show their morphological analysis each year respectively. It is necessary because a seasonal variation in morphological characteristics can have a place.”
Response: We appreciate the Points and suggestions. As mentioned in the Material and Methods section, all analyses were performed only with female mosquitoes. Table 2 shows the number of analyzed specimens as well as other relevant information. The mosquitoes were identified using the taxonomic key by Consoli & Lourenço-de-Oliveira (1994). Mosquitoes were collected monthly for 24 months, and when possible were randomly selected to avoid using siblings, which would introduce a bias to the analysis. For more details on the number of collected mosquitoes please see Medeiros-Sousa et al. 2017.
Medeiros-Sousa, A.R.; Fernandes, A.; Ceretti-Junior, W.; Wilke, A.B.B.; Marrelli, M.T. Mosquitoes in urban green spaces: using an island biogeographic approach to identify drivers of species richness and composition. Sci. Rep. 2017, 7, 17826.
Furthermore, please note that wing shape is considered a conserved trait and would not change significantly over the period of the study, please see Henry et al. 2010 for more details.
Henry, A.; Thongsripong, P.; Fonseca-Gonzalez, I.; Jaramillo-Ocampo, N.; Dujardin, J.-P.P. Wing shape of dengue vectors from around the world. Infect. Genet. Evol. 2010, 10, 207–214.
We modified the sentence to: “To avoid using sibling specimens, the specimens were randomly selected, and different collection techniques were used: manual 12 V aspirators [47], CO2-baited CDC light traps and Shannon traps [48].”
Point 2: “The identified morphological changes may also be related to the water temperature of larval stage development. High and low water temperatures have different effects to the larval development, its size, and, accordingly, can effect to the morphological parameters of the imago”.
Response: We appreciate the Point and agree with the importance of further discussing this topic. Indeed, it has been described by previous studies (Stephens, C.R.; Juliano, S.A. Wing shape as an indicator of larval rearing conditions for Aedes albopictus and Aedes aegypti (Diptera: Culicidae). J. Med. Entomol. 2012, 49, 927–938.). That changes regarding water temperature and physicochemical parameters of water can affect the size of the wings. However, wing shape is considered a conserved trait and would not be affected by punctual environmental variations, please see Henry, et al 2010. Furthermore, our objective was not to focus on specific breeding sites or to investigate the possible variations and potential drivers according to the intrinsic characteristics of breeding sites. Our goal was to investigate how invasive (Ae. albopictus) and native (Ae. scapularis) populations react to similar selective pressure in urban areas. (Henry, A.; Thongsripong, P.; Fonseca-Gonzalez, I.; Jaramillo-Ocampo, N.; Dujardin, J.-P. Wing shape of dengue vectors from around the world. Infect. Genet. Evol. 2010, 10, 207–214.)
Point 3: “Also, it was interesting to indicate the ratio of species in the sympatric zones, whether there is competition between the two species and its affect to the morphological variations of the two species Aedes albopictus and Ae. scapularis”.
Response: We thank you for the Point. We agree with the importance of determining how the competition is affecting the population dynamics of these mosquito vector populations. However, it is important to keep in mind that the objective of this study was not to investigate the possible existence and the level of population structure of the species due to competition. That would have to be done using a completely different experimental setup and cannot be included in this present manuscript. Correspondingly, here we focused on the hypothesis that under similar environmental and temporal conditions, populations of Ae. scapularis and Ae. albopictus, native and invasive mosquito species, respectively, in Brazil, have been affected differently by urbanization processes.
Reviewer 2 Report
The authors of this study intend to demonstrate that urbanization processes affect two different Aedes species differently. They analyzed wing shape variation to justify their hypothesis.
In general, I do not think that the results presented were adequate enough, to make any conclusive statements about the hypothesis being described in the introduction. (Either the aim and the discussion should be modified to reflect directly the immediate message of the current results OR more results are required to allow the reader to make the link between morphometric results with urbanization process)
- All hypotheses testing in the results were only relating to intraspecies geographic variation.
- Interspecies comparisons were only descriptive. It would help to actually do hypothesis tests on between species variation, perhaps differences in coefficient of variation? Also only three of the seven populations have both species sampled. What would happen if comparisons were only performed for those three populations? How do you justify making comparisons between the two species when there are non-shared locations?
- There was no attempt to make any statistical link with quantitative measures of urbanization, apart from just descriptions in the discussion. It would help to provide more quantitative data in the results that suggest different locations have undergone different levels of urbanization OR more information on the collection locations. This may give a bit more reason for discussing urbanization process in the Discussion.
In more detail:
Please review some of the citations. Some examples:
- in the first sentence where you stated "... has the greatest number of species... ", the citation number 1 does not explicitly provide numerical comparison between genera.
- The link in citation 2 does not seem to work.
Lines 128 to 130 and Figure 2:
In figure 2, are the figures swapped? (A) is Ae. albopictus, while (B) is Ae. scapularis. If this is the case, then in the text the reported mean for ALV and STD populations would be swapped?
Lines 138 to 140:
Allometry was removed from subsequent analysis? Do you mean size effect on shape were removed by some protocol (e.g. superimposition) ?
Line 143:
How do you quantify variation and how is Ae. albopictus having 'moderate variation'? By looking at the percentage variation explained?
Lines 143 to 146:
If ALV differed the most from all others, which appear quite clear in Figure 3A, how is it that BLM differed the most from STD? ALV does not overlap with BLM, but STD does.
Figure 3:
Why is there a number '3' in the middle left side of the figure? What do the outlines of the populations represent?
Lines 165:
Could the separation between ANH and BLM population be explained by genetic components? The same could be asked for the statement the authors made in lines 169 to 171.
Line 178:
ALV misspelt as AFV
Table 4 and 5:
When looking at common populations only (ANH, BLM and PQR), some of the percentages were higher in Ae. albopictus than Ae. scapularis. Do the percentages have variance? E.g. How do we know that 55.5% statistically significantly greater than 45% (BLM vs ANH)
Figure 5:
Explain what does the light blue and dark blue wireframe represent (e.g. light blue, mean shape? ...)
How do you know that the nett variation of landmarks is greater in Ae. albopictus? Could a single large variation in Ae. scapularis (e.g. landmark 10) contribute more variation than the smaller changes in multiple landmarks in Ae. albopictus?
Discussion section:
As mentioned before, without some quantitative links made in the results... the urbanization narrative requires readers to assume that shape trends can only be explain by environmental factor and then assume that urbanization is the only environmental factor.
Lack of heterogeneity in one species should be tested directly by comparing variance (or coefficient of variation) rather than by looking just at the face value of the magnitude of intraspecies differences.
Lines 253 to 264 (just an opinion of the flow):
Initially, I thought it was going off in a tangent when the authors talked about another paper which appear to explain findings from another paper. It might make it less confusing if the results of this study are highlighted first and then the links to the other papers made after that.
Author Response
Dear Reviewer
Thank you very much for reviewing our manuscript and for the suggestions, which have enabled us to greatly improve the text.
We have made the changes to the manuscript to address the points, and we hope the article is now suitable for publication in Insects.
Sincerely,
Mauro Marrelli
Reviewer 2 Points
Point 1: “In general, I do not think that the results presented were adequate enough, to make any conclusive statements about the hypothesis being described in the introduction. (Either the aim and the discussion should be modified to reflect directly the immediate message of the current results OR more results are required to allow the reader to make the link between morphometric results with urbanization process)”.
Response: We thank the reviewer for raising these issues. We agree with your observation and toned down the conclusions and modified the discussion of the manuscript to be better aligned with the main hypothesis. The intraspecific analyses showed a clear difference in structural levels of wing shape variation between the populations of the two species. Our discussion was also based on the fact that these two species have different colonization histories, with Ae. albopictus, an invasive species that was only detected in São Paulo no more than 30 years ago, which might explain its homogenous variation on wing shape in the populations.
Point 2: “All hypotheses testing in the results were only relating to intraspecies geographic variation”
Response: We thank you for this Point. In our opinion, our approach made it possible to assess how invasive (Ae. albopictus) and native (Ae. scapularis) populations react to similar selective pressure in urban areas. This is especially important since there are not so many studies comparing how invasive species can adapt to new areas nor how native species cope with increased urbanization.
Point 3: “Interspecies comparisons were only descriptive. It would help to actually do hypothesis tests on between species variation, perhaps differences in coefficient of variation?”
Response: We are grateful for the reviewer's Points and suggestions. The points raised are very relevant and were taken into account for the improvement of the manuscript. We performed additional analyses of Procrustes distance to highlight the different levels of wing shape variation between Ae. albopictus and Ae. scapularis populations, please see Supplementary Material, Table S3, Table S4, Figure S1.
Point 4: “Also only three of the seven populations have both species sampled. What would happen if comparisons were only performed for those three populations?”
Response: We appreciate the suggestion. The initial analyses were performed considering those three populations, and the results were similar to the ones obtained when all populations were analyzed. Ae. albopictus populations yielded homogeneous wing patterns, whereas Ae. scapularis populations yielded higher levels of variation. However, we managed to collect enough mosquito specimens in different areas and decided to include them in this project. By including more populations our conclusions are based on a higher number of collection sites and specimens of both species. We added this information to the end of the Results section.
Point 5: “How do you justify making comparisons between the two species when there are non-shared locations?”
Response: We appreciate the point the reviewer has raised. The fact of only three out of seven populations were sympatric was not a limiting factor in our study. Even though Ae. albopictus and Ae. scapularis were only sympatric for 3 locations it did to represent a limitation since as discussed above, wing shape variation is a conserved trait and not directly affected by short term environmental changes.
Point 6: “There was no attempt to make any statistical link with quantitative measures of urbanization, apart from just descriptions in the discussion. It would help to provide more quantitative data in the results that suggest different locations have undergone different levels of urbanization OR more information on the collection locations. This may give a bit more reason for discussing urbanization process in the Discussion”
Response: We thank you for the Point. The text was revised and we have added more information to Table 1 and in the Discussion section. As showed by Medeiros-Sousa et al [22] and Multini et al [34], the loss of vegetal cover (a widely accepted proxy for urbanization) is an important driver of variation in populations of vector mosquito species. However, please keep in mind that identifying the exact drivers responsible for driving wing shape variation was not the objective of our study. Further studies with different experimental designs are needed to shed light on this topic.
Point 6: “Lines 128 to 130 and Figure 2: In figure 2, are the figures swapped? (A) is Ae. albopictus, while (B) is Ae. scapularis. If this is the case, then in the text the reported mean for ALV and STD populations would be swapped?”
Response: We thank you for the observation. The figure has been corrected
Point 7: “in the first sentence where you stated "... has the greatest number of species... ", the citation number 1 does not explicitly provide numerical comparison between genera. The link in citation 2 does not seem to work”.
Response: We agree with the observation. The sentence that referred to the species "the greatest number" has been deleted. As for the link to citation 2, the text was modified to improve clarity.
Point 8: “Allometry was removed from subsequent analysis? Do you mean size effect on shape were removed by some protocol (e.g. superimposition)?”
Response: We appreciate the observation and have included in the text more detailed information on how the allometry was removed. The manuscript now reads: “ To assess the effect of wing size on wing shape (allometry) we performed a multivariated regression of the Procrustes coordinates against Centroid Size using a permutation test with 10,000 randomizations in MorphoJ 1.02”
Point 9: “How do you quantify variation and how is Ae. albopictus having 'moderate variation'? By looking at the percentage variation explained?”
Response: Thank you for pointing out the need for a more detailed description. The terms "moderate" and "high" were replaced by "heterogeneous" and "homogeneous", respectively. As cited above, we added more analyses (Procrustes distance, Table S3, Table S4, Figure S1) to better support our results and conclusions.
Point 10: “If ALV differed the most from all others, which appear quite clear in Figure 3A, how is it that BLM differed the most from STD? ALV does not overlap with BLM, but STD does.”
Response: We thank you for your Point and for pointing out this imprecision. We corrected the sentence: BLM differed the most from ALV.
Point 11: “Why is there a number '3' in the middle left side of the figure? What do the outlines of the populations represent?”
Response: We appreciate the observation. The number "3" was an error in the production of the figure. We have corrected it.
Point 12: “Could the separation between ANH and BLM population be explained by genetic components? The same could be asked for the statement the authors made in lines 169 to 171.”
Response: We appreciate the question. The separation between ANH and BLM may be an indication of genetic components; however, further analyses would be required to test this hypothesis. As for the results shown in lines 169 -171, since they are the result of the Centroid Size analysis, it is not possible to attribute genetic components to what has been observed. Wing size variation is considered strictly phenotypical and cannot be related to gene expression.
Point 13: “ALV misspelt as AFV”
Response: We thank you for the observation. The text was corrected
Point 14: “Table 4 and 5:
When looking at common populations only (ANH, BLM and PQR), some of the percentages were higher in Ae. albopictus than Ae. scapularis. Do the percentages have variance? E.g. How do we know that 55.5% statistically significantly greater than 45% (BLM vs ANH)”
Response: We appreciate the question. Although, in the sympatric populations, there is a greater variation in Ae. albopictus than in Ae. scapularis the objective of this study was not to assess interspecific competition between the two species. This hypothesis is relevant and interesting but requires a different experimental design from the one used in this study. We have added more information to the methods section.
“Cross-validated reclassification was carried out for each Ae. albopictus and Ae. scapularis specimen for all populations with MorphoJ 1.05 to evaluate the degree of dissimilarity between samples. Each mosquito was reclassified according to its wing similarity to the average shape of each group using based on Mahalanobis distances to test the accuracy of morphometric analyses.”
The percentages do not have variance. They represent the exact number of specimens that were correctly identified (according to the hypothesis tested – in this case, collection point source location). We used percentages to facilitate the interpretation of the results. Differences in reclassification values are due to natural variations in wing shape patterns between populations. When populations (of same species) share similar wing shape patters but one of these populations has higher levels of wing shape variability and the other population has lower levels of wing shape variability, the first will yield higher values in the pairwise cross-validated reclassification test, whereas the latter will yield lower values.
Point 15: “Figure 5: Explain what does the light blue and dark blue wireframe represent (e.g. light blue, mean shape? ...)”
Response: We appreciate the Point, and have included more information in the legend of the figure: “Superimposed wireframe graphs of Aedes scapularis (A) and Aedes albopictus (B). The light blue lines represent the medium wing shape variation for all populations and the dark blue lines represent the shape variance based on CV1.”
Point 16: “How do you know that the nett variation of landmarks is greater in Ae. albopictus? Could a single large variation in Ae. scapularis (e.g. landmark 10) contribute more variation than the smaller changes in multiple landmarks in Ae. albopictus?”
Response: We appreciate the question. However, we have to consider the total variation instead of focusing on the unique variation of each anatomical landmark. The contribution of each landmark can vary and is not necessarily associated with more or less variation. On some occasions, more subtle variations in a given anatomical landmark can be more meaningful if that landmark is more conserved, whereas, in a less conserved anatomical landmark, higher levels of variation can be diluted due to background noise caused by the higher levels of variation.
Point 17: “As mentioned before, without some quantitative links made in the results... the urbanization narrative requires readers to assume that shape trends can only be explain by environmental factor and then assume that urbanization is the only environmental factor.”
Response: We thank you for the Point. The term "urbanization" was used as a set of factors that determine a change in the habitat, as evidenced in: "Urbanization results in a range of environmental impacts, including an increase in temperature, loss of habitat, pollution, and deforestation [53]. Anthropogenic changes to the environment often benefit exotic mosquito species that can exploit the available resources and thrive in human-modified habitats [54].
Point 18: “Lack of heterogeneity in one species should be tested directly by comparing variance (or coefficient of variation) rather than by looking just at the face value of the magnitude of intraspecies differences.”
Response: We appreciate the Point and agree with the point raised by the reviewer. As mentioned before, we performed additional analyses that confirmed that one species has less wing shape variation (Ae. albopictus) and the other has higher levels of wing shape variation (Ae. scapularis) (Table S3, Table S4, Figure S1).
Point 19: “Lines 253 to 264 (just an opinion of the flow):
Initially, I thought it was going off in a tangent when the authors talked about another paper which appear to explain findings from another paper. It might make it less confusing if the results of this study are highlighted first and then the links to the other papers made after that”
Response: We appreciate the suggestion and modified the paragraph to improve the flow of information.
Reviewer 3 Report
Reviewer comments
Using wing geometric morphometrics method, the authors studied the microgeographic population structure of Aedes albopictus and Ae. Scapularis in San Paulo, Brazil, to demonstrate the different degree of intraspecific variation of these two mosquito groups across different city parks. The results reveal the influence of human impacts on wing shape patterns in the populations of these two mosquito species, in order to provide necessary guidance for controlling vector-borne disease transmission by these insects. In generally, the manuscript is well written and the reviewer think it can be accepted for publication in Insects after addressing the following minor revisions:
- Table 1 should be constrained in one page, not be separated in two pages, and Table 3 as well.
- Figures should be improved with higher resolution and larger font.
- The References should be formatted with necessary space between the number (year or volume or page number) and punctuation.
Author Response
Dear Reviewer
Thank you very much for reviewing our manuscript and for the suggestions, which have enabled us to greatly improve the text.
We have made the changes to the manuscript to address the points, and we hope the article is now suitable for publication in Insects.
Sincerely,
Mauro Marrelli
Reviewer 3 Points:
Point 1:“Table 1 should be constrained in one page, not be separated in two pages, and Table 3 as well.”
Response: We thank you for the Point. We corrected the format in the new version.
Point 2: “Figures should be improved with higher resolution and larger font.“
Response: We thank you for the Point. The figures indicated by the reviewer have been improved in quality as requested.
Point 3: “The References should be formatted with necessary space between the number (year or volume or page number) and punctuation “
Response: We thank you for pointing out this issue. References have now been formatted according to Insects guidelines.
Round 2
Reviewer 1 Report
The article by Rafael Oliveira-Christe, André Barretto Bruno Wilke and Mauro Toledo Marrelli "Microgeographic wing-shape variation in Aedes albopictus and Aedes scapularis (Diptera: Culicidae) populations" has been edited by the authors according to my comments. The authors did a great job and answered all my questions, which satisfied me quite well. This article can be published in the journal "Insects".
Author Response
We appreciate the reviewer's suggestions and are thankful for his kind words.
Reviewer 2 Report
There has been some improvement, but some of the changes are not clear.
Please provide line number where changes have been made in text in responses to comments.
- line 123 - "... multivariate regression..."
- line 151-153, perhaps it should read something like "...Ae. scapularis wing shape is more heterogeneous than Ae. aegypti" i.e. relative difference in heterogeneity. Otherwise the reader will be left wondering how you classify something as homogeneous or heterogeneous by itself.
- Also line 151-153, if the intention is to say one is more heterogeneous than the other, please provide some values for comparison and/or a statistical test (e.g. test differences in variances OR coefficient of variation)
- line 157, "... ANH had the most differences..." reads oddly, do you mean ANH was most differentiated from all other populations... ?
- lines 160-161 - this sentence does not read right and difficult to follow. Instead of citing multiple tables and figure for multiple results... incorporate the citation to each individual results in the previous sentences.
- Lines 166-167, sentence does not make sense "... which varied more than the corresponding figures..." (What figures?)
- Lines 206-209, please detail in the text how you know Ae. albopictus has more variation in landmarks than Ae. scapularis. It is not clear by just looking at Figure 5. Did you compute total variation and showed that Ae. albopictus is higher?
- Lines 210-213, please provide results for the populations that had both mosquito species sampled (at least in the supplementary)
- Lines 235-237, the word '... is ...' should be removed
- Lines 276-287, referring to my comments in the first review, it's not clear what has been changed here.
- Line 284, do you mean 'than' instead of 'then'
Author Response
line 123 - "... multivariate regression..."
Response: Thank you for pointing out this typo. we have corrected it.
line 151-153, perhaps it should read something like "...Ae. scapularis wing shape is more heterogeneous than Ae. aegypti" i.e. relative difference in heterogeneity. Otherwise the reader will be left wondering how you classify something as homogeneous or heterogeneous by itself.
Response: We agree with the editor and have rewritten the sentence as requested. We assumed the reviewer meant Ae. albopictus instead of Ae. aegypti. The sentence now reads “The results of the CVA differed with Ae. scapularis wing shape being more heterogeneous than Ae. albopictus. Among the former, the ALV population differed the most from all the others: it was completely segregated from the ANH and BLM populations and overlapped with PQR and STD only slightly. The Ae. scapularis populations that differed the most from each other were BLM and ALV (Figure 3A).”
Also line 151-153, if the intention is to say one is more heterogeneous than the other, please provide some values for comparison and/or a statistical test (e.g. test differences in variances OR coefficient of variation)
Response: We have rephrased this sentence as suggested above. However, please keep in mind that we have provided the Procrustes Distance values to support higher levels of wing shape variation in Ae. scapularis when compared to Ae. albopictus. Please refer to Figure S1 and Table S4.
line 157, "... ANH had the most differences..." reads oddly, do you mean ANH was most differentiated from all other populations... ?
Response: Thank you for pointing out the need to improve this sentence. We have rewritten it to improve clarity. It now reads “Among the Ae. albopictus populations, ANH had higher levels of wing shape variation, with just a minor overlap with the PQR population and no similarities with the PRV population. The BLM population had the greatest degree of overlap with all the other populations, and the SHL population also overlapped with all the other populations, although to a lesser extent with the ANH population (Figure 3B).”
lines 160-161 - this sentence does not read right and difficult to follow. Instead of citing multiple tables and figure for multiple results... incorporate the citation to each individual results in the previous sentences.
Response: Thank you for pointing out the need to improve this sentence. We have rewritten it to improve clarity. It now reads “This result is in agreement with the ones obtained in the UPGMA phenogram based on Procrustes distances of Ae. scapularis and Ae. albopictus populations (Table S3, Table S4, Figure S1).”
The tables and figure citations are all referent to the sentence in which they have been cited. For this reason, they were kept at the end of the sentence.
Lines 166-167, sentence does not make sense "... which varied more than the corresponding figures..." (What figures?)
Response: Thank you for pointing out the need to improve the clarity of this sentence. We have rewritten it. It now reads “Aedes scapularis populations yielded higher levels of variation for the Mahalanobis distances, 1.6815 to 3.4416, when compared to Ae. albopictus populations (1.6163 to 2.6631). All Ae. scapularis comparisons were statistically significant (P<0.05), the only exception being ANH vs. BLM (P=0.1846) (Table 2). A similar result was obtained for Ae. albopictus, in which except for SHL vs. PRV, all the other comparisons were statistically significant (P<0.05) (Table 3).”
Lines 206-209, please detail in the text how you know Ae. albopictus has more variation in landmarks than Ae. scapularis. It is not clear by just looking at Figure 5. Did you compute total variation and showed that Ae. albopictus is higher?
Response: We agree with the point raised by the reviewer and have rewritten the sentence to address the point he/she has raised. The sentence now reads “The wireframe graph shows the levels of variation in landmarks for Ae. albopictus and Ae. scapularis. In the latter, landmarks 1, 2, 10, 11 and 13 varied more than the remaining landmarks whereas in the former all the landmarks showed some degree of variation, although landmarks 1, 2, 3, 8, 9, 10, 12, 14 and 15 varied more (Figure 5).”
Lines 210-213, please provide results for the populations that had both mosquito species sampled (at least in the supplementary)
Response: We agree with the reviewer on the importance of focusing on the population dynamics and microevolution of sympatric populations. However, please consider that analyzing sympatric populations was never in the scope of this study, and, in our opinion, would not contribute to test the current hypothesis of our manuscript.
Lines 235-237, the word '... is ...' should be removed
Response: Thank you for pointing that out. “is” has been deleted as instructed.
Lines 276-287, referring to my comments in the first review, it's not clear what has been changed here.
Response: As suggested by the reviewer we have included the Procrustes distances for both species analyzed in our study. By using the Procrustes distances values we were able to produce the necessary values requested by the reviewer allowing us to compare the levels of variation based on a coefficient of variation (ranging from 0 to 1).
Line 284, do you mean 'than' instead of 'then'
Response: Thank you for pointing out this typo. We have corrected it accordingly.